# Bridging Generalization Gap of Heterogeneous Federated Clients Using Generative Models

**Ziru Niu**
School of Computing Technologies
RMIT University
Melbourne, VIC, Australia
ziru.niu@student.rmit.edu.au

**Hai Dong** *
School of Computing Technologies
RMIT University
Melbourne, VIC, Australia
hai.dong@rmit.edu.au

**A. K. Qin**
Department of Computing Technologies
Swinburne University of Technology
Hawthorn, VIC, Australia
kqin@swin.edu.au

## Abstract

Federated Learning (FL) is a privacy-preserving machine learning framework facilitating collaborative training across distributed clients. However, its performance is often compromised by data heterogeneity among participants, which can result in local models with limited generalization capability. Traditional model-homogeneous approaches address this issue primarily by regularizing local training procedures or dynamically adjusting client weights during aggregation. Nevertheless, these methods become unsuitable in scenarios involving clients with heterogeneous model architectures. In this paper, we propose a model-heterogeneous FL framework that enhances clients' generalization performance on unseen data without relying on parameter aggregation. Instead of model parameters, clients share feature distribution statistics (mean and covariance) with the server. Then each client trains a variational transposed convolutional neural network using Gaussian latent variables sampled from these distributions, and use it to generate synthetic data. By fine-tuning local models with the synthetic data, clients achieve significant improvement of generalization ability. Experimental results demonstrate that our approach not only attains higher generalization accuracy compared to existing model-heterogeneous FL frameworks, but also reduces communication costs and memory consumption. The experiment code is available at: https://github.com/ZiruNiu0/FedVTC.

## 1 Introduction

The prevalence of the Internet of Things (IoT) propels federated learning (FL) (McMahan et al., 2017) as a widespread technique to process the dispersed data among end clients while ensuring data privacy. Under the regulation of a central server, clients collaboratively train a global model without revealing any personal data, and exchange the model parameters with the server. Abstaining from data transmission, FL markedly alleviates communication overhead and the risk of data leakage. However, the inherent data heterogeneity of IoT clients presents significant challenges to FL. Firstly, clients comprise edge devices distributed across various geographical locations, which naturally collect non-identically-independent data (Abeysekara et al., 2023). Secondly, single clients usually collect real-time data with frequent distribution shifts in practical scenarios, such as weather forecasting (Mclaughlin & Su, 2024; Reddy et al., 2023) and healthcare (Saha et al., 2017).

---

*Corresponding author.

To overcome both inter-client and intra-client data heterogeneities, improving the *generalization* ability (i.e., the ability to predict unseen data correctly (Sun et al., 2024)) of clients has become a vital matter in FL (Huang et al., 2024). Existing solutions mainly focus on two aspects, including regularization and weight modification. Regularization methods propose to debias the individual training process by imposing a regularization term on the local objective function of clients (Dinh et al., 2020; Li et al., 2021b; 2020b; Smith et al., 2017; Karimireddy et al., 2020; Acar et al., 2021; Gao et al., 2022; Li et al., 2019). Weight modification methods aim to alleviate the negative impact of biased clients on the global model, including assigning lower weights to biased entities in aggregation (Pillutla et al., 2022; Cao et al., 2020; Mu et al., 2024; Li et al., 2020a; Tahmasebian et al., 2022) or reducing the possibility of selecting these clients (Jee Cho et al., 2022; Li et al., 2022; Niu et al., 2024; Zhang et al., 2023; Lai et al., 2021). As a result, the global model maintains significant robustness against the heterogeneous client data. Correspondingly, clients can improve their generalization performance by incorporating the enhanced global model into their local models.

Despite the outstanding progress in addressing data heterogeneity, the aforementioned works are developed based on the assumption that all clients share the same model architecture. In practice, clients are likely to obtain personalized models with diverse and irrelevant architectures (Huang et al., 2024), referred to as *model-heterogeneous* FL (see Appendix A for details), making traditional aggregation-based methods prohibitive. In this case, clients cannot learn generalized information through parameter sharing, which necessitates alternative strategies for enhancing the generalization of local models with differing architectures.

Several attempts have been made to improve the generalization of model-heterogeneous FL through knowledge distillation (Li & Wang, 2019; Itahara et al., 2023; Cho et al., 2022; Nguyen et al., 2023; Cheng et al., 2021; Sun & Lyu, 2021; Sattler et al., 2021). These works utilize a shared public dataset and enable clients to exchange knowledge by comparing each other's predictions on the dataset. However, the prerequisite of **a public dataset** severely restricts the deployment of these works, as a public dataset is usually not available in real-world circumstances (Zhu et al., 2021).

To address this concern, data-free model-heterogeneous FL methods have been proposed, which can mainly be divided into two categories. The first category employs generated feature representations for knowledge distillation (Zhu et al., 2021; Luo et al., 2021), eliminating the dependence on public datasets. However, unlike raw samples, knowledge distillation on feature space merely regularizes the head classifier of a local model, while leaving the backbone feature extractor unmodified. The second category involves circumventing knowledge distillation and enabling clients to communicate through alternative messages, such as mean feature representations (Tan et al., 2022; Zhang et al., 2024) and a mutual proxy model (Wang et al., 2024). Although (Tan et al., 2022; Zhang et al., 2024) regularizes local feature extractors by enforcing them to derive unbiased feature representations, the biased head classifiers are neglected in these works, which significantly limits the generalization improvement. For (Wang et al., 2024), training and transmitting an overparameterized proxy model (Wang et al., 2024) will result in substantial communication and memory overhead.

To overcome the disadvantages of prior methods, we expect a comprehensive FL framework that can increase the generalization of model-heterogeneous clients without relying on any public dataset, while maintaining communication and memory efficiency. To this end, we propose **FedVTC** (Federated Learning with Variational Transposed Convolution), a model-heterogeneous FL framework that uses synthetic data to fine-tune local models, thus improving generalization without relying on a public dataset. FedVTC achieves this by enabling each client to generate synthetic samples using a variational transposed convolutional (VTC) neural network. The VTC model takes low-dimensional Gaussian latent variables as input and produces corresponding synthetic samples by upsampling the latent variables. Afterwards, clients fine-tune the local models with the synthetic samples to improve the generalization ability. In communication, clients only exchange the local mean and covariance with the server to mitigate the bias of local feature distributions, leading to substantial communication cost reduction. Compared with representation-based knowledge distillation (Zhu et al., 2021; Luo et al., 2021) that only regularizes the head classifier, fine-tuning with synthetic images can debias both the feature extractor and the classifier in a local model, thereby potentially achieving higher generalization performance. In addition, FedVTC enables clients to train the VTC model and local models alternately to avoid extra memory consumption.

Similar to a variational autoencoder (VAE), a VTC model is initially trained by maximizing the evidence lower bound (ELBO) of local data (Kingma & Welling, 2013). Moreover, to strengthen

the robustness of VTC against variant input latent variables, we incorporate a Distribution Matching (DM) loss (Zhao et al., 2023) into the objective function of VTC for regularization. This strategy helps FedVTC produce synthetic samples with better training quality, which notably improves the performance of the fine-tuned local models. The contributions of this paper are summarized as follows:

- We propose FedVTC, a model-heterogeneous FL framework based on variational transposed convolution. FedVTC uses synthetic data to fine-tune local models for better generalization, eradicating the dependence on pre-existing public datasets.

- We design a novel objective function to train the VTC model, which encompasses the standard negative ELBO loss and a DM loss. The DM loss regularizes the training procedure of VTC, enabling VTC to consistently generate high-quality synthetic samples with varying input latent variables.

- We conduct extensive experiments to validate the effectiveness of FedVTC. Experiment results show that FedVTC obtains higher generalization accuracy than existing model-heterogeneous FL frameworks over MNIST, CIFAR10, CIFAR100 and Tiny-ImageNet datasets, as well as lower or equivalent communication costs and memory consumption.

## 2 LITERATURE REVIEW

### 2.1 IMPROVING THE GENERALIZATION OF MODEL-HOMOGENEOUS FEDERATED LEARNING

The endeavors of the state-of-the-art to enhance the generalization of model-homogeneous FL can mainly be divided into two categories, which are regularization and weight modification. **Regularization** approaches aim to debias the local training procedure by adding a regularization term onto the clients' local objective functions. For instance, (Dinh et al., 2020; Li et al., 2021b; 2020b; Karimireddy et al., 2020; Acar et al., 2021; Gao et al., 2022) prevent client models from overfitting to local optima by restricting the Euclidean distance between local and global parameters. Besides, (Smith et al., 2017) employs a relationship matrix to enforce clients with higher similarities to learn close parameter updates, and (Li et al., 2019) alleviates the bias of local training using ridge regression.

**Weight modification** methods enhance the generalization of the global model by deducting the proportion of biased clients participating in global training. Subsequently, clients can also reinforce the generalization ability by incorporating the well-generalized global model into their local models. On one hand, (Pillutla et al., 2022; Cao et al., 2020; Mu et al., 2024; Li et al., 2020a; Tahmasebian et al., 2022) replace weighted average with dynamic weighting strategies, and assign lower weights to biased, anomalous and malicious parameter updates in aggregation. On the other hand, (Jee Cho et al., 2022; Li et al., 2022; Niu et al., 2024; Zhang et al., 2023; Lai et al., 2021) design some heuristic functions to evaluate the importance of each client, such as training loss and cosine similarity. Moreover, pFedGen (Le et al., 2024) enables unseen clients to generate representation vectors using a feature extractor trained by previous clients, and only interacts with clients with the smallest representation distance. Consequently, these works develop a selective client sampling strategy that selects unimportant entities with lower possibilities to mitigate the negative effect of biased clients. Weight modification methods enhance the generalization of the global model by deducting the proportion of biased clients participating in global training.

Although these works have achieved remarkable progress in improving FL generalization, they are developed based on the assumption that all clients share a universal model architecture. For model-heterogeneous FL, where clients exhibit diverse model architectures, the aforementioned works become infeasible.

### 2.2 MODEL-HETEROGENEOUS FEDERATED LEARNING

The most commonly used approach to improve the generalization of model-heterogeneous FL is **knowledge distillation** (Li & Wang, 2019; Itahara et al., 2023; Cho et al., 2022; Nguyen et al., 2023; Cheng et al., 2021; Sun & Lyu, 2021; Sattler et al., 2021). With the presence of a public dataset, clients are able to acquire unbiased global knowledge by learning from each other's predictions on the dataset. However, the prerequisite of a public dataset can be a critical bottleneck for these methods, as such a dataset is not usually available in practice.

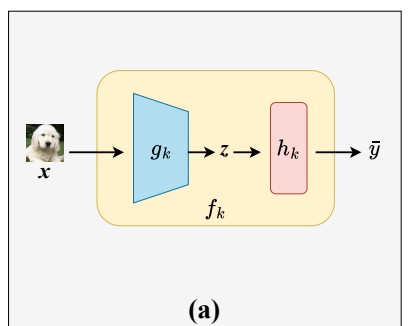 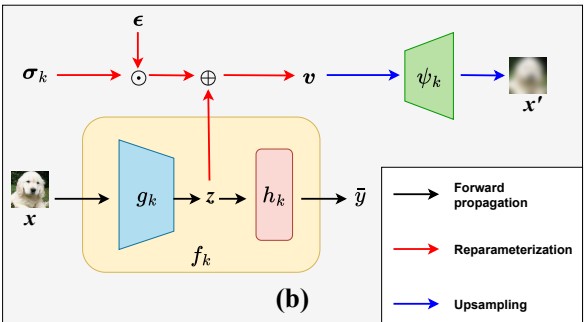

Figure 1: A comparison between: **(a)**: a standard FL client and **(b)**: a client in FedVTC. In (b), "⊕" and "⊙" respectively standard for element-wise addition and product.

To overcome the dependence on a public dataset, (Zhu et al., 2021; Luo et al., 2021) propose to use generative models, such as a generative adversarial network (GAN) and a multivariate Gaussian sampler, to generate feauture representations for knowledge distillation. However, unlike raw samples that progress throuth all layers, feature representations merely go through the head classifier of a neural network. Consequently, the distillation scheme in (Zhu et al., 2021; Luo et al., 2021) only debiases the classifier of local models, which significantly limits the generalization improvement. Additionally, instead of knowledge distillation, (Tan et al., 2022; Zhang et al., 2024; Wang et al., 2024) allows clients to exchange model-architecture-independent messages for generalization, such as the mean of feature representations (Tan et al., 2022; Wang et al., 2024) and a common proxy model (Wang et al., 2024). For one thing, representation sharing (Tan et al., 2022; Wang et al., 2024) regularizes the backbone feature extractors in local models by enforcing clients to derive unbiased feature representations, while neglecting the head classifiers. For another thing, training and transmitting a proxy model (Wang et al., 2024) might cause undesired communication and memory costs.

## 2.3 FEDERATED TRAINING OF GENERATIVE MODELS

Most existing FL schemes for training a generative model have several practical limitations. For instance, (Stanley Jothiraj & Mashhadi, 2024) and (Peng et al., 2025) propose to train a global diffusion model in FL networks. However, they require a public dataset to facilitate knowledge distillation among client models, which is not usually accessible. (Zhang et al., 2022) trains a global generative adversarial network (GAN) by maximizing the discrepancy between client models. Clients use the downloaded GAN to produce training samples from unseen classes, thereby enhancing the performance of zero-shot prediction. However, the transmission of client models might cause overwhelming communication overhead. (Shi et al., 2024) utilizes a contrastive language-image pretraining (CLIP) model to guide clients' local training for better few/zero-shot performance, while a pre-existing CLIP model is not usually available.

## 3 METHODOLOGY

### 3.1 PRELIMINARIES

Suppose a network consisting of a central server and a set of clients $\mathcal{K} = \{1, 2, ..., K\}$ with non-iid local datasets $\{D_1, ..., D_K\}$. Each client $k$ ($1 \le k \le K$) obtains a local model $f_k : \mathbb{R}^d \to \mathbb{R}$, which can be decomposed into a backbone feature extractor $g_k : \mathbb{R}^d \to \mathbb{R}^p$ and a head classifier $h_k : \mathbb{R}^p \to \mathbb{R}$. In formula, $f_k = h_k \circ g_k$. Let $\{(\boldsymbol{x}_i, y_i)\}_{1 \le i \le n_k}$ denote the collection of data in the local dataset $D_k$ with a total of $n_k = |D_k|$ samples. As shown in Figure 1(a), for each sample $\boldsymbol{x} \in \mathbb{R}^d$ with the corresponding label $y \in \{1, 2, ......, C\}$, $f_k$ takes $\boldsymbol{x}$ as input and makes a prediction $\bar{y}$, where $C$ is the total number of all possible classes. Firstly, $\boldsymbol{x}$ is transformed to a latent variable $\boldsymbol{z} \in \mathbb{R}^p$ using the feature extractor $g_k$ (i.e., $\boldsymbol{z} = g_k(\boldsymbol{x})$). Secondly, $\boldsymbol{z}$ is forwarded to the classifier $h_k$ to produce a prediction $\bar{y} = h_k(\boldsymbol{z})$.

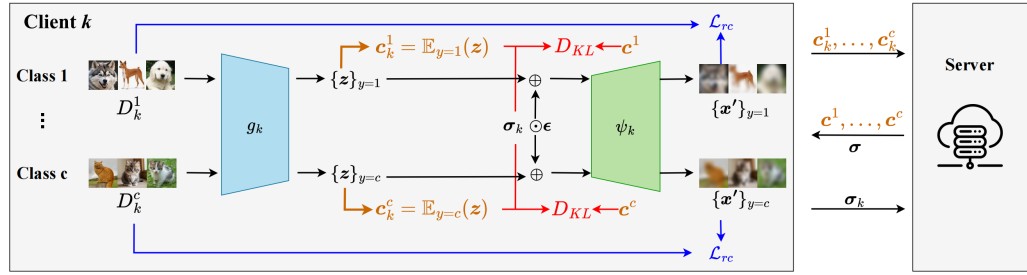

Figure 2: VTC is trained with loss function $\mathcal{L}_e = \mathcal{L}_{rc} + D_{KL}$. $\mathcal{L}_{rc}$ (blue) is the reconstruction loss between the original samples and the generated samples, and $D_{KL}$ (red) is the KL-divergence between the distributions of the local and global latent variables.

The objective of model-heterogeneous FL is to find the optimal set of local models $\{f_1^*, ......, f_K^*\}$ minimizing the local empirical loss, that is:

$$f_1^*, ......, f_K^* = \underset{f_1, ..., f_K}{\arg\min} \frac{1}{K} \sum_{k=1}^{K} \frac{1}{|D_k|} \sum_{(\boldsymbol{x}, y) \in D_k} \mathcal{L}(y, f_k(\boldsymbol{x})) \tag{1}$$

$\mathcal{L}(y, f_k(\boldsymbol{x}))$ is the classification loss (e.g., cross-entropy) of model $f_k$ on the sample $(\boldsymbol{x}, y)$. In addition, to evaluate generalization performance, we run each $f_k$ on $D_0$, which is a test dataset consisting of unseen samples (Sun et al., 2024), that is, $\forall k, D_0 \cap D_k = \varnothing$.

## 3.2 VARIATIONAL TRANSPOSED CONVOLUTION

As Figure 1(b) shows, aside from the fundamental local model $f_k$, FedVTC introduces a variational transposed convolution (VTC) model $\psi_k : \mathbb{R}^p \to \mathbb{R}^d$ to each client $k$. VTC is an upsampling network that takes a latent variable $\boldsymbol{z} \in \mathbb{R}^p$ as input to produce an enlarged data sample $\boldsymbol{x}' \in \mathbb{R}^d$ (Dumoulin & Visin, 2016). Similar to a VAE (Kingma & Welling, 2013), for each sample $\boldsymbol{x}$ with the corresponding $\boldsymbol{z}$, client $k$ samples a random variable $\boldsymbol{v}$ from the distribution $\mathcal{N}(\boldsymbol{v}|\boldsymbol{z}, \Sigma_k)$, and forwards $\boldsymbol{v}$ to $\psi_k$ to derive $\boldsymbol{x}'$ (blue arrow in Figure 1(b)). The covariance matrix $\Sigma_k \in \mathbb{R}^{p \times p}$ can be learned via the gradient method. To make $\Sigma_k$ differentiable, FedVTC applies the well-known reparametrization trick as per VAE and lets $\boldsymbol{v} = \boldsymbol{z} + \boldsymbol{\sigma}_k \odot \boldsymbol{\epsilon}$ (red arrow in Figure 1(b)). $\boldsymbol{\epsilon} \in \mathbb{R}^p$ is a random Gaussian noise with distribution $\boldsymbol{\epsilon} \sim \mathcal{N}(\boldsymbol{\epsilon}|\boldsymbol{0}, \boldsymbol{I})$, $\boldsymbol{\sigma}_k = [\sigma_1, \sigma_2, ......, \sigma_p]^\top$ indicates the reparameterized standard deviation (SD), and "$\odot$" represents element-wise product. In this case, for each entry in $\Sigma_k$, we have $(\Sigma_k)_{ii} = \sigma_i^2$ and $(\Sigma_k)_{ij} = 0$ for any $i \neq j$, and each $\sigma_i$ ($1 \leq i \leq p$) can be learned through gradient descent.

Following the design of VAE (Kingma & Welling, 2013), VTC is trained by maximizing the **evidence lower bound (ELBO)**, which is calculated as:

$$\text{ELBO} = log\, p_{\psi_k}(\boldsymbol{x}'|\boldsymbol{z}, \boldsymbol{\sigma_k}) - D_{KL}(q_{g_k}(\boldsymbol{z}|\boldsymbol{x})||p(\boldsymbol{z})) \tag{2}$$

$log\, p_{\psi_k}(\boldsymbol{x}'|\boldsymbol{z}, \boldsymbol{\sigma_k})$ represents how $\psi_k$ can generate real samples based on the latent variable $\boldsymbol{z} + \boldsymbol{\sigma}_k \odot \boldsymbol{\epsilon}$. According to (Kingma & Welling, 2013), this term can be well approximated by minimizing the empirical reconstruction loss $\mathcal{L}_{rc}(\boldsymbol{x}', \boldsymbol{x}) = \|\boldsymbol{x}' - \boldsymbol{x}\|_2^2$ as Figure 2 shows. The negative Kullback–Leibler (KL)-divergence $-D_{KL}(q_{g_k}(\boldsymbol{z}|\boldsymbol{x})||p(\boldsymbol{z}))$ measures how $q_{g_k}(\boldsymbol{z}|\boldsymbol{x})$ (the distribution of $\boldsymbol{z}$ learned by the feature extractor $g_k$) approaches the real distribution $p(\boldsymbol{z})$. Following (Kingma & Welling, 2013), we can assume $p(\boldsymbol{z}) \sim \mathcal{N}(\boldsymbol{\mu}, \boldsymbol{I})$, with $\boldsymbol{\mu}$ denoting the unbiased mean of $\boldsymbol{z}$ which is usually $\boldsymbol{0}$ by default. In FL settings, we follow (Tan et al., 2022) and estimate $\boldsymbol{\mu}$ using $\boldsymbol{c}^y$, which is the global **prototype** of class $y$. As Figure 2 shows, a global prototype $\boldsymbol{c}^y = \frac{1}{|\mathcal{K}_y|} \sum_{k \in \mathcal{K}_y} \boldsymbol{c}_k^y$ is calculated as the expectation of all local prototype $\boldsymbol{c}_k^y$ for all $k \in \mathcal{K}_y$, with $\mathcal{K}_y \subset \mathcal{K}$ indicating the set of all clients containing data of class $y$. A local prototype of client $k$ is the mean of the local feature representations, i.e. $\boldsymbol{c}_k^y = \frac{1}{|D_k^y|} \sum_{(\boldsymbol{x}, y) \in D_k^y} g_k(\boldsymbol{x})$, where $D_k^y \subset D_k$ indicates the collection of all samples of class $y$ in local dataset $D_k$. In consequence, for two Gaussian distributions $q_{g_k}(\boldsymbol{x}'|\boldsymbol{z}, \boldsymbol{\sigma_k})$ and $p(\boldsymbol{z})$, the KL-divergence can be expressed analytically:

$$D_{KL}(q_{g_k}(\boldsymbol{x}'|\boldsymbol{z}, \boldsymbol{\sigma_k})||p(\boldsymbol{z})) = \frac{1}{2}\big[(\boldsymbol{z} - \boldsymbol{c}^y)^T(\boldsymbol{z} - \boldsymbol{c}^y) + tr(\Sigma_k) - p - log|\Sigma_k|\big] \tag{3}$$

Therefore, we derive a negative ELBO loss $\mathcal{L}_e$ to optimize $g_k$, $\psi_k$ and $\boldsymbol{\sigma_k}$:

$$\begin{aligned} \mathcal{L}_e &= \sum_{y \in \mathcal{Y}_k} \frac{1}{|D_k^y|} \sum_{i=1}^{|D_k^y|} \mathcal{L}_{rc}(\boldsymbol{x}_i', \boldsymbol{x}_i) + D_{KL}(q_{g_k}(\boldsymbol{z_i}|\boldsymbol{x_i})||p(\boldsymbol{z})) \\ &= \sum_{y \in \mathcal{Y}_k} \frac{1}{|D_k^y|} \sum_{i=1}^{|D_k^y|} \|\boldsymbol{x}_i' - \boldsymbol{x}_i\|_2^2 + \frac{1}{2}\big[(\boldsymbol{z}_i - \boldsymbol{c}^y)^T(\boldsymbol{z}_i - \boldsymbol{c}^y) + tr(\Sigma_k) - p - log|\Sigma_k|\big] \end{aligned} \tag{4}$$

$\mathcal{Y}_k = \{y : D_k^y \neq \varnothing, 1 \leq y \leq C\}$ indicates all possible classes from client $k$'s data.

As Figure 2 shows, to mitigate the bias of the local distribution $\mathcal{N}(\boldsymbol{z}, \Sigma_k)$, each client $k$ will update the local class-wise prototypes $\{\boldsymbol{c}_k^y\}_{y \in \mathcal{Y}_k}$ and the SD $\boldsymbol{\sigma}_k$ to the server. In response, the server obtains the global prototypes $\{\boldsymbol{c}^y\}$ and SD $\boldsymbol{\sigma}$ through aggregation, and sends $\{\boldsymbol{c}^y\}, \boldsymbol{\sigma}$ back to clients.

## 3.3 REGULARIZING VTC FOR BETTER GENERALIZATION

Owing to the randomness of the VTC process, $\psi_k$ is likely to generate variant synthetic samples $\{\boldsymbol{x}'\}$ with unknown training quality. To address this issue, we propose to regularize the training procedure of VTC with a distribution matching (DM) loss $\mathcal{L}_{dm}$:

$$\mathcal{L}_{dm} = \sum_{y \in \mathcal{Y}_k} \frac{1}{|D_k^y|} \sum_{i=1}^{|D_k^y|} \|g_k(\boldsymbol{x}_i') - \boldsymbol{c}^y\|_2^2 = \sum_{y \in \mathcal{Y}_k} \frac{1}{|D_k^y|} \sum_{i=1}^{|D_k^y|} \|g_k(\psi_k(\boldsymbol{v}_i)) - \boldsymbol{c}^y\|_2^2 \tag{5}$$

This loss enforces $\psi_k$ to generate high-quality samples in which $g_k$ can acquire a feature distribution close to the unbiased global distribution (Zhao et al., 2023). Subsequently, the local model $f_k$ will learn unbiased parameter updates from the synthetic data, resulting in better generalization performance.

Overall, the loss function $\mathcal{L}_{tc}$ to train a VTC model is defined as:

$$\mathcal{L}_{tc} = \mathcal{L}_e + \lambda \mathcal{L}_{dm} \tag{6}$$

$\lambda > 0$ is the coefficient of regularization.

## 3.4 FEDVTC OVERVIEW

The comprehensive FedVTC framework is presented in Algorithm 1. Specifically, for each iteration $t$, every active client $k$ regularly trains the local model $f_k$ as in traditional FL. In addition, $k$ trains $\psi_k$ and $\boldsymbol{\sigma}_k$ using loss function $\mathcal{L}_{tc}$ (Equation 6). Afterwards, $k$ uploads the local prototypes and SD to the server for aggregation. The server averages the received results and returns the global prototype and SD to clients.

For memory efficiency, FedVTC applies an alternating strategy in VTC training. For line 12 in Algorithm 1, we first freeze $\psi_k$ and $\boldsymbol{\sigma}_k$ and incorporate the step $g_k \leftarrow g_k - \eta \nabla_{g_k} \mathcal{L}_{tc}$ into the standard training of $f_k$. In this case, line 11 in Algorithm 1 can be written as $f_k \leftarrow f_k - \eta \nabla_{f_k}(\mathcal{L} + \mathcal{L}_{tc})$ (with $g_k$ included in $f_k$). Then we freeze $f_k$ and optimize $\psi_k$ and $\boldsymbol{\sigma}_k$ with $\mathcal{L}_{tc}$. As a result, FedVTC will not extend the maximum memory usage, as $\psi_k$ and $\boldsymbol{\sigma}_k$ usually require less memory space than $f_k$ for training. In contrast, if we optimize $f_k$, $\psi_k$ and $\boldsymbol{\sigma}_k$ simultaneously, the maximum memory usage will be enlarged to their accumulated memory usage.

Once FL is complete, every client $k$ uploads the local VTC $\psi_k$ to the server, then the server derives a global VTC $\psi$ through aggregation, and broadcasts $\psi$ along with $\{\boldsymbol{c}^1, ..., \boldsymbol{c}^C\}$, $\boldsymbol{\sigma}_k$ to all clients. In order to aggregate VTC parameters, FedVTC lets all $\psi_k$'s have the same architecture. For each $\boldsymbol{c}^y$ $(y \in \{1, ..., C\})$, every client $k$ samples $S$ latent variables from distribution $\mathcal{N}(\boldsymbol{c}^y, \Sigma)$, and generates the corresponding synthetic samples by forwarding these variables to $\psi$. $\Sigma \in \mathbb{R}^{p \times p}$ is a diagonal

---

**Algorithm 1** FedVTC

---

**Require:** FL iterations $T$, fine-tuning rounds $\mathcal{T}$, local epochs $E$, clients $\mathcal{K} = \{1, ..., K\}$, local models $\{f_1, ..., f_K\}$, datasets $\{D_1, ..., D_K\}$, local VTCs $\{\psi_1, ..., \psi_K\}$, local SDs $\{\boldsymbol{\sigma}_1, ..., \boldsymbol{\sigma}_K\}$, initial SD $\boldsymbol{\sigma}$, initial prototypes $\{\boldsymbol{c}^1, ..., \boldsymbol{c}^C\}$ learning rate $\eta$, number of generated samples $S$.
1: **for** $t = 1, 2, ..., T$ **do**
2:     **server does**:
3:         randomly sample a set of participating clients $\mathcal{K}^t \subset \mathcal{K}$.
4:         **for** $y \in \{1, ..., C\}$:
5:             $\mathcal{K}_y^t \leftarrow \mathcal{K}^t \cap \mathcal{K}_y$.
6:             broadcast $\boldsymbol{c}^y$ to clients in $\mathcal{K}_y^t$.
7:         broadcast $\boldsymbol{\sigma}$ to clients in $\mathcal{K}^t$.
8:     **every client** $k \in \mathcal{K}^t$ **does**:
9:         $\boldsymbol{\sigma}_k \leftarrow \boldsymbol{\sigma}$.
10:        **for** epochs $1, ..., E$:
11:           $f_k \leftarrow f_k - \eta \nabla_{f_k} \mathcal{L}$.                                  ▷ Update $f_k$ with SGD.
12:           $g_k \leftarrow g_k - \eta \nabla_{g_k} \mathcal{L}_{tc}, \psi_k \leftarrow \psi_k - \eta \nabla_{\psi_k} \mathcal{L}_{tc}, \boldsymbol{\sigma}_k \leftarrow \boldsymbol{\sigma}_k - \eta \nabla_{\boldsymbol{\sigma}_k} \mathcal{L}_{tc}$ .
13:        **for** $y \in \mathcal{Y}_k$:
14:           $\boldsymbol{c}_k^y \leftarrow \frac{1}{|D_k^y|} \sum_{\boldsymbol{x} \in D_k^y} g_k(\boldsymbol{x})$.                      ▷ Compute local prototype.
15:           upload $\boldsymbol{c}_k^y$ to the server.
16:        upload $\boldsymbol{\sigma}_k$ to the server.
17:     **server does:**
18:         **for** $y \in \{1, ..., C\}$:
19:             $\boldsymbol{c}^y \leftarrow \frac{1}{|\mathcal{K}_y^t|} \sum_{k \in \mathcal{K}_y^t} \boldsymbol{c}_k^y$.                     ▷ Update global prototype.
20:         $\boldsymbol{\sigma} \leftarrow \frac{1}{|\mathcal{K}^t|} \sum_{k \in \mathcal{K}^t} \boldsymbol{\sigma}_k$.                         ▷ Update global SD.
21: **end for**
22: **every client** $k \in \mathcal{K}$ **does**:
23:     upload $\psi_k$ to the server.
24: **server does:**
25:     $\psi \leftarrow \frac{1}{K} \sum_{k=1}^K \psi_k$.                            ▷ Obtain global VTC.
26:     broadcast $\psi, \{\boldsymbol{c}^1, ..., \boldsymbol{c}^C\}$ and $\boldsymbol{\sigma}$ to every client $k \in \mathcal{K}$.
27: **for** every $k \in \mathcal{K}$ **do**
28:     create synthetic dataset $D_k'$ with $S$ samples using $\psi, \boldsymbol{\sigma}$ and $\{\boldsymbol{c}^1, ..., \boldsymbol{c}^C\}$.
29:     **for** rounds $1, ..., \mathcal{T}$:
30:         $\mathcal{L}' \leftarrow \frac{1}{S} \sum_{i=1}^S (y_i', f(\boldsymbol{x}_i'))$.
31:         $f_k \leftarrow f_k - \eta \nabla_{f_k} \mathcal{L}'$.                 ▷ Fine-tuning with the synthetic dataset.
32: **end for**
33: **return** $w^t$

---

matrix with $\Sigma_{ii}$ equal to the squared $i-$th element in $\boldsymbol{\sigma}$. Afterwards, each client fine-tunes its local model by running classification tasks on the total $S \times C$ synthetic samples. In this stage, all local fine-tuning processes run in isolation, and no communication occurs between clients and the server.

One noteworthy point in FedVTC is that, clients only need to upload the local VTC models once instead of every FL iteration (see line 23 in Algorithm 1). In the experiment, we compare the performance of FedVTC in cases of singular VTC transmission and per-round VTC transmission. The results show that these two modules exhibit marginal differences in terms of accuracy (see Table 3). Therefore, in the ultimate FedVTC framework, local VTCs are only communicated once for communication efficiency.

## 4 EXPERIMENTS

### 4.1 EXPERIMENT SETUP

**Datasets.** We evaluate FedVTC on the following datasets: MNIST (LeCun, 1998) contains a training set with 60,000 samples and a validation set with 10,000 samples from 10 classes. CIFAR10

| | *Dir* | FedProto | FedTGP | FedGen | CCVR | FedType | pFedAFM | **FedVTC** |
|---|---|---|---|---|---|---|---|---|
| MNIST | 0.1 | 83.4±0.4 | 85.4±0.5 | 81.1±1.0 | 85.5±0.4 | 83.9±0.3 | 85.6 ± 0.5 | **88.7**±0.7 |
| | 1.0 | 86.1±0.3 | 86.8±0.4 | 81.9±0.3 | 86.2±0.3 | 86.7±0.5 | 87.2 ± 0.4 | **90.1**±0.1 |
| CIFAR10 | 0.1 | 39.3±0.7 | 40.9±0.2 | 39.1±0.2 | 39.3±0.2 | 39.7±0.3 | 38.9 ± 0.9 | **46.9**±0.6 |
| | 1.0 | 41.1±0.2 | 42.1±0.7 | 39.7±0.5 | 41.0±0.1 | 40.2±0.6 | 40.5 ± 0.5 | **51.7**±0.7 |
| CIFAR100 | 0.1 | 26.2±0.2 | 29.2±0.3 | 27.3±0.8 | 27.6±0.9 | 31.3±1.5 | 32.2 ± 1.0 | **36.3**±0.8 |
| | 1.0 | 30.4±0.2 | 31.8±0.5 | 30.8±0.3 | 29.3±0.4 | 34.4±0.6 | 34.9 ± 0.6 | **40.4**±0.3 |
| Tiny-ImageNet | 0.1 | 23.7±0.3 | 24.9±0.8 | 23.6±0.1 | 24.1±0.3 | 24.8±0.4 | 24.2 ± 0.8 | **30.2**±0.3 |
| | 1.0 | 26.4±0.2 | 27.2±0.3 | 26.1±0.1 | 26.6±0.8 | 31.6±0.4 | 27.3 ± 0.2 | **35.8**±0.5 |

Table 1: The average generalization accuracy (in %) of each client's local model on the global validation dataset (with mean ± SD).

and CIFAR100 (Krizhevsky, 2009) contain a training set with 50,000 samples and a validation set of 10,000 samples from 10 and 100 classes. Tiny-ImageNet (Chrabaszcz et al., 2017) contains a training set with 100,000 samples and a validation set with 10,000 samples from 200 classes.

**Comparative methods.** We compare FedVTC with five state-of-the-art model-heterogeneous FL frameworks. **1. FedProto** (Tan et al., 2022) enables clients to regularize the local feature extractors using class-wise global prototypes. **2. FedTGP** (Zhang et al., 2024) extends the design of Fed-Proto by optimizing the decision boundary between inter-class prototypes using gradient descent. **3. FedGen** (Zhu et al., 2021) and **4. CCVR** (Luo et al., 2021) utilizes virtual feature representations to fine-tune the classifiers of clients, with FedGen using a generative adversarial network (GAN) and CCVR using a Gaussian distribution to generate the representations. **5. FedType** (Wang et al., 2024) allows clients to collaboratively train a proxy model using conformal prediction and enhance generalization by distilling knowledge through the proxy model. **6. pFedAFM (Yi et al., 2025)** clients employ a proxy feature embedding model for knowledge distillation.

**System implementation.** We simulate a virtual network consisting of $K = 100$ clients with a participation rate of 0.1 (i.e. $|\mathcal{K}^t| = 10$). Both the server and all clients operate on one NVIDIA Geforce RTX 4070 GPU with 12 GB of RAM space. For heterogeneous client models, we follow the settings in (Zhang et al., 2024; Wang et al., 2024), where clients are divided into five uniform clusters with five model architectures: ResNet-18, ResNet-34, ResNet-50, ResNet-101 and ResNet-152 (He et al., 2016). For non-iid local data distribution, we follow (Luo et al., 2021) and distribute the training sets of MNIST, CIFAR10, CIFAR100, and Tiny-ImageNet unevenly among clients using two Dirichlet (*Dir*) distributions with parameters 0.1 and 1.0. The experiment platform is implemented with PyTorch 2.0 (Paszke et al., 2019). The VTC model has a simple architecture with four transposed convolution layers (see Appendix B.2).

**Parameter settings.** Following (Tan et al., 2022; Wang et al., 2024), the total number of iterations is set to $T = 100$. For FedVTC, the number of additional fine-tuning rounds is set to $\mathcal{T} = 5$. For fairness, we also let the baselines train extra $\mathcal{T}$ rounds with full client participation so that all methods have the same training rounds. The learning rate $\eta$ is set to 1e-4 for MNIST (Luo et al., 2021) and 0.01 for others (Tan et al., 2022). The local epoch $E$ is set to five, and the batch size is set to 16 universally (Wang et al., 2024). For fairness, the number of synthetic samples $S$ is set to 500 for MNIST and CIFAR10, and 1000 for CIFAR100 and Tiny-ImageNet, which is identical to the amount of generated representations in (Luo et al., 2021). We set the regularization parameter as $\lambda = 0.1$ (see Appendix B.4). The settings of $p$ and $d$ can be found in Appendix B.1.

## 4.2 EXPERIMENT RESULTS

**Generalization accuracy.** We evaluate each client model with the global validation sets in the aforementioned four datasets for generalization accuracy. Each method runs for three independent trials, and the average results are recorded. As shown in Table 1, FedVTC consistently outperforms the comparative methods across all datasets and distributions, demonstrating a remarkable generalization ability over non-iid data. As a supplementary, the graphs depicting accuracy per iteration are included in Appendix B.3.

|  | FedProto | FedTGP | FedGen | CCVR | FedType | pFedAFM | **FedVTC** |
|---|---|---|---|---|---|---|---|
| MNIST | 0.94 | 0.94 | 13.15 | 46.61 | 12.37 | 1.94 | **0.70** |
| CIFAR10 | 3.93 | 3.93 | 54.58 | 807.46 | 37.45 | 5.88 | **2.90** |
| CIFAR100 | 39.32 | 39.32 | 91.07 | 825.16 | 72.85 | 41.27 | **26.49** |
| Tiny-ImageNet | 78.64 | 78.64 | 844.82 | 923.46 | 179.61 | 147.10 | **52.71** |

Table 2: The total volume of data transmission (in GB).

|  |  | MNIST | | CIFAR10 | | CIFAR100 | | Tiny-ImageNet | |
|---|---|---|---|---|---|---|---|---|---|
|  |  | $Dir(0.1)$ | $Dir(1.0)$ | $Dir(0.1)$ | $Dir(1.0)$ | $Dir(0.1)$ | $Dir(1.0)$ | $Dir(0.1)$ | $Dir(1.0)$ |
| * | ① | 88.7(0.7) | 90.1(0.1) | **46.9**(0.6) | 51.7(0.7) | 36.3(0.8) | **40.4**(0.3) | 30.2(0.3) | 35.8(0.5) |
|  | ② | **89.7**(0.2) | **92.3**(0.3) | 45.4(0.3) | **54.3**(0.5) | **37.7**(0.4) | 39.6(0.6) | **31.8**(0.7) | **38.0**(1.0) |
| ** | ① | **0.70** | | **2.90** | | 26.49 | | 52.71 | |
|  | ② | 5.63 | | 8.77 | | 32.36 | | 123.22 | |

Table 3: The (*)generalization accuracy (%) and (**)communication cost (GB) of ①**FedVTC with singular VTC transmission** and ②**FedVTC with regular VTC transmission per round**.

|  | MNIST | | CIFAR10 | | CIFAR100 | | Tiny-ImageNet | |
|---|---|---|---|---|---|---|---|---|
|  | $Dir(0.1)$ | $Dir(1.0)$ | $Dir(0.1)$ | $Dir(1.0)$ | $Dir(0.1)$ | $Dir(1.0)$ | $Dir(0.1)$ | $Dir(1.0)$ |
| $\mathcal{L}_e$ | 87.2(0.4) | 89.9(0.3) | 39.7(0.5) | 45.4(0.4) | 28.9(0.5) | 35.2(0.2) | 28.1(0.1) | 33.4(0.3) |
| $\mathcal{L}_{tc}$ | **88.7**(0.7) | **90.1**(0.1) | **46.9**(0.6) | **51.7**(0.7) | **36.3**(0.8) | **40.4**(0.3) | **30.2**(0.3) | **35.8**(0.5) |

Table 4: The generalization accuracy of two training modules of FedVTC. The first module uses the standard $\mathcal{L}_e$ loss, and the second module uses the regularized $\mathcal{L}_{tc} = \mathcal{L}_e + \lambda\mathcal{L}_{dm}$ loss.

**Communication efficiency.** As shown in Table 2, FedVTC obtains the least overall communication cost among all methods. Specifically, for CCVR, clients share the local prototypes and covariates with the server. Although transmitting the $p-$dimensional prototypes introduces marginal communication cost, this cost grows exponentially for transmitting the covariance matrices of a $p \times p$ dimension. For FedGen and FedType, the server respectively broadcasts a generator (2-layer perception (Zhu et al., 2021)) and a proxy model (ResNet-18 (Wang et al., 2024)) to clients, while these overparametrized models usually consume massive communication resources in transmission. In comparison, FedProto and FedTPG only transmit prototypes with significantly less communication costs. Although FedVTC transmits additional messages other than prototypes, including SDs and VTC models, it still reduces the total communication costs compared with FedProto and FedTGP, as no information is transmitted in the phase of fine-tuning.

**Memory requirement.** We calculate the required memory space of each method by accumulating the magnitudes of parameter weights, gradients and activations involved in training (Pfeiffer et al., 2023). As depicted in Figure 3, compared with other methods, FedVTC accounts for an equivalent or less memory space across all model architectures. We attribute this to the strategy of alternately training local models and VTC models in FedVTC. In contrast, FedType, which simultaneously trains local models and the proxy model, demands a much higher memory space for training.

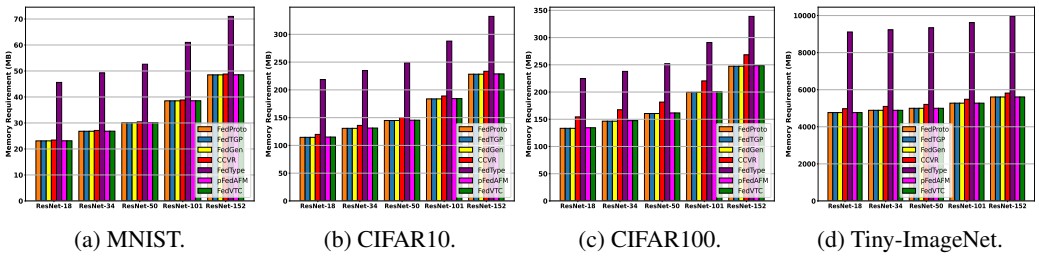

(a) MNIST.      (b) CIFAR10.      (c) CIFAR100.      (d) Tiny-ImageNet.

Figure 3: FedVTC (green) has an equivalent or less memory requirement compared with others.

**Ablation study.** We compare the performance of FedVTC with two modules of VTC transmission, including **singular** (TC models are communicated only once) and **regular** (TC models are communicated every round) transmissions. As shown in Table 3, the generalization accuracy of singular VTC transmission is slightly lower (sometimes even higher) than regular VTC transmission, while the latter causes much higher communication cost than the former. Therefore, we apply the singular VTC transmission strategy in the ultimate FedVTC framework for communication efficiency. Furthermore, we compare two training modules of FedVTC, where the VTC is trained with $\mathcal{L}_e$ and $\mathcal{L}_{tc} = \mathcal{L}_e + \lambda \mathcal{L}_{dm}$ respectively. As shown in Table 4, the DM-based regularization effectively increases the generalization performance of FedVTC in all scenarios.

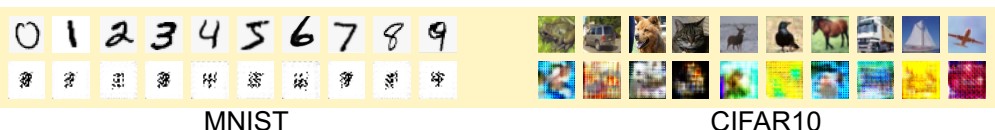

MNIST                                                       CIFAR10

Figure 4: A fraction of raw images (top) and synthetic images (bottom) on MNIST and CIFAR10.

**Privacy preservation.** FedVTC transmits prototypes, SDs and VTC models, which brings no additional privacy concerns according to (Luo et al., 2021; Tan et al., 2022). The reason is that, it is almost impossible to recover the raw data from prototypes and SDs without accessing the raw representations. As Figure 4 shows, the VTC-generated images diverge drastically from real images, making it extremely difficult to capture any sensitive information from these images.

## 5 CONCLUSION

This paper proposes FedVTC, a model-heterogeneous FL framework based on variational transposed convolution (VTC). By fine-tuning local models with VTC-generated samples, FedVTC effectively overcomes the low-generalization bottleneck of FL clients in model-heterogeneous settings without relying on any public datasets. In the future, we aim to consolidate FedVTC with the early-stopping technique (Niu et al., 2024) to mitigate the additional computation cost of training a VTC. We also plan to amplify the adaptability of FedVTC to data from unseen classes with techniques like zero-shot learning (Chen et al., 2024) and model diffusion (Croitoru et al., 2023).

## ACKNOWLEDGEMENTS

This work is funded by the Australian Research Council under Grant No. DP220101823, DP200102611, and LP180100114.

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

## A DEFINITION OF MODEL HETEROGENEITY

To our best knowledge, the definitions of "model-heterogeneous federated learning" in the state-of-the-art can be divided into the following two categories.

## A.1 SUB-MODEL-BASED MODEL HETEROGENEITY

Sub-model-based model heterogeneity, also named *partial* model heterogeneity (Ye et al., 2023), is developed based on the assumption that all local models are a subset of the global model. Specifically, the server preserves a global model and employs the typical dropout technique (Caldas et al., 2018) to extract a fraction of the global model's parameters, i.e., a *sub-model*, and allocates the sub-models to clients. For example, (Caldas et al., 2018) generates sub-models by randomly pruning neurons in the global model. (Horváth et al., 2021) and (Diao et al., 2021) consistently extract the left-most neurons of the global model to form sub-models. (Alam et al., 2022) dynamically extracts sub-models using a sliding window across the neurons in the global model. (Jiang et al., 2023; Li et al., 2021a; Jiang et al., 2022) selectively prune the unimportant neurons in the global model with the least heuristic scores (like parameter norms). (Kim et al., 2023) vertically extracts sub-models by pruning the top layers in the global model.

In these works, although clients have different model architectures due to different dropout strategies, the parameters between clients are usually co-dependent, and can be aggregated using sub-model aggregation as shown in Figure 5.

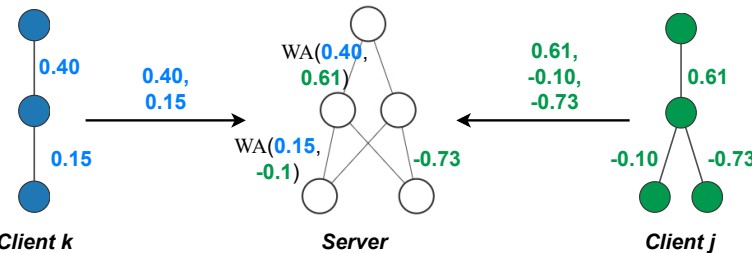

Figure 5: The sub-model aggregation scheme for sub-model-based model-heterogeneous FL ("WA" stands for weighted average).

## A.2 COMPLETE MODEL HETEROGENEITY

In completely model-heterogeneous FL (shown in Figure 6), the server no longer maintains a global model, and clients obtain isolated local models with diverse, independent, and irrelevant architectures (Ye et al., 2023). In this scenario, clients can no longer acquire generalized parameter updates through model aggregation. As a replacement, clients apply alternative strategies to learn generalization information, such as sharing knowledge over a public dataset or transmitting parameter-independent messages, as discussed in Section 2.2.

**Without ambiguity, in this paper, the definition of "model heterogeneity" refers to complete model heterogeneity (Appendix A.2) where all aggregation (e.g., standard aggregation and sub-model aggregation) methods become prohibitive. Correspondingly, the goal of FedVTC is to improve the generalization performance of clients in the settings of complete model heterogeneity without parameter aggregation involved.**

## B EXPERIMENT DETAILS

### B.1 FOUNDAMENTAL SETTINGS

The fundamental experimental settings are summarized in Table 5.

### B.2 STRUCTURE OF THE TRANSPOSED CONVOLUTION MODEL

The detailed information on the structure of the VTC model $\psi$ is listed in Table 6.

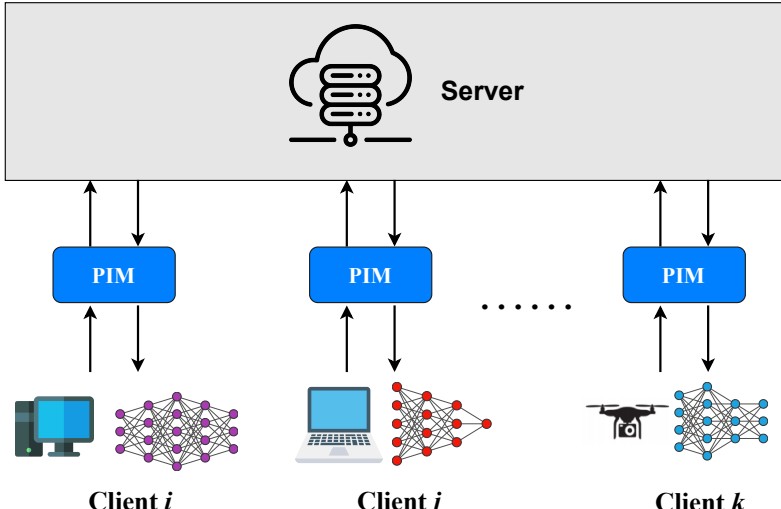

Figure 6: In complete model-heterogeneous FL, the model architectures between clients are independent, irrelevant, and cannot be aggregated altogether. Alternatively, clients communicate some Parameter-Independent Messages (PIM) with the server for knowledge sharing, such as prototypes and feature covariates.

| | Channel | Input dimension $d$ | Latent dimension $p$ | Class $C$ | Initial SD $\sigma$ |
|---|---|---|---|---|---|
| MNIST | 1 | $1 \times 28 \times 28$ | 980 | 10 | |
| CIFAR10 | 3 | $3 \times 32 \times 32$ | 4096 | 10 | $\mathbf{1} \in \mathbb{R}^p$ |
| CIFAR100 | 3 | $3 \times 32 \times 32$ | 4096 | 100 | |
| Tiny-ImageNet | 3 | $3 \times 64 \times 64$ | 12768 | 200 | |

Table 5: Fundamental experimental settings.

### B.3 ACCURACY GRAPHS

The graphs reflecting the generalization accuracy against the FL iterations are shown in Figure 7.

### B.4 HYPERPARAMETER TUNING

As stated previously, the number of synthetic samples per class $S$ is set to 500 for MNIST and 1000 for CIFAR10, CIFAR100 and Tiny-ImageNet following (Luo et al., 2021) for fairness. Despite this, we additionally tune $S$ on the MNIST dataset to study the impact of different $S$ values. As shown in Figure 8a, when $S$ is too small, the generalization accuracy is low as the effect of fine-tuning is weak. When $S$ is too large, many redundant synthetic samples will be generated, which may degrade the effectiveness of fine-tuning.

In addition, we evaluated the coffecient $\lambda$ of the DM loss (Eq. (5)). As shown in Figure 8b, it turns out that $\lambda = 0.1$ is the optimal setting based on the empirical results.

| MNIST | | | |
|---|---|---|---|
| **Index** | **Components** | **Input shape** | **Output shape** |
| 1 | TransposeConv2d(kernel=3, padding=1, stride=1)
BatchNorm2d()
LeakyReLU(0.01) | $20 \times 7 \times 7$ | $16 \times 7 \times 7$ |
| 2 | TransposeConv2d(kernel=4, padding=1, stride=2)
BatchNorm2d()
LeakyReLU(0.01) | $16 \times 7 \times 7$ | $32 \times 14 \times 14$ |
| 3 | TransposeConv2d(kernel=3, padding=1, stride=1)
BatchNorm2d()
LeakyReLU(0.01) | $32 \times 14 \times 14$ | $32 \times 14 \times 14$ |
| 4 | TransposeConv2d(kernel=4, padding=1, stride=2)
BatchNorm2d()
Sigmoid() | $32 \times 14 \times 14$ | $1 \times 28 \times 28$ |

| CIFAR10 & CIFAR100 | | | |
|---|---|---|---|
| **Index** | **Components** | **Input shape** | **Output shape** |
| 1 | TransposeConv2d(kernel=3, padding=1, stride=1)
BatchNorm2d()
LeakyReLU(0.01) | $64 \times 8 \times 8$ | $32 \times 8 \times 8$ |
| 2 | TransposeConv2d(kernel=4, padding=1, stride=2)
BatchNorm2d()
LeakyReLU(0.01) | $32 \times 8 \times 8$ | $64 \times 16 \times 16$ |
| 3 | TransposeConv2d(kernel=3, padding=1, stride=1)
BatchNorm2d()
LeakyReLU(0.01) | $64 \times 16 \times 16$ | $64 \times 16 \times 16$ |
| 4 | TransposeConv2d(kernel=4, padding=1, stride=2)
BatchNorm2d()
Sigmoid() | $64 \times 16 \times 16$ | $3 \times 32 \times 32$ |

| Tiny-ImageNet | | | |
|---|---|---|---|
| **Index** | **Components** | **Input shape** | **Output shape** |
| 1 | TransposeConv2d(kernel=3, padding=1, stride=1)
BatchNorm2d()
LeakyReLU(0.01) | $128 \times 16 \times 16$ | $64 \times 16 \times 16$ |
| 2 | TransposeConv2d(kernel=4, padding=1, stride=2)
BatchNorm2d()
LeakyReLU(0.01) | $64 \times 16 \times 16$ | $64 \times 32 \times 32$ |
| 3 | TransposeConv2d(kernel=3, padding=1, stride=1)
BatchNorm2d()
LeakyReLU(0.01) | $64 \times 32 \times 32$ | $64 \times 32 \times 32$ |
| 4 | TransposeConv2d(kernel=4, padding=1, stride=2)
BatchNorm2d()
Sigmoid() | $64 \times 32 \times 32$ | $3 \times 64 \times 64$ |

Table 6: Structural information on the transposed convolutional neural network.

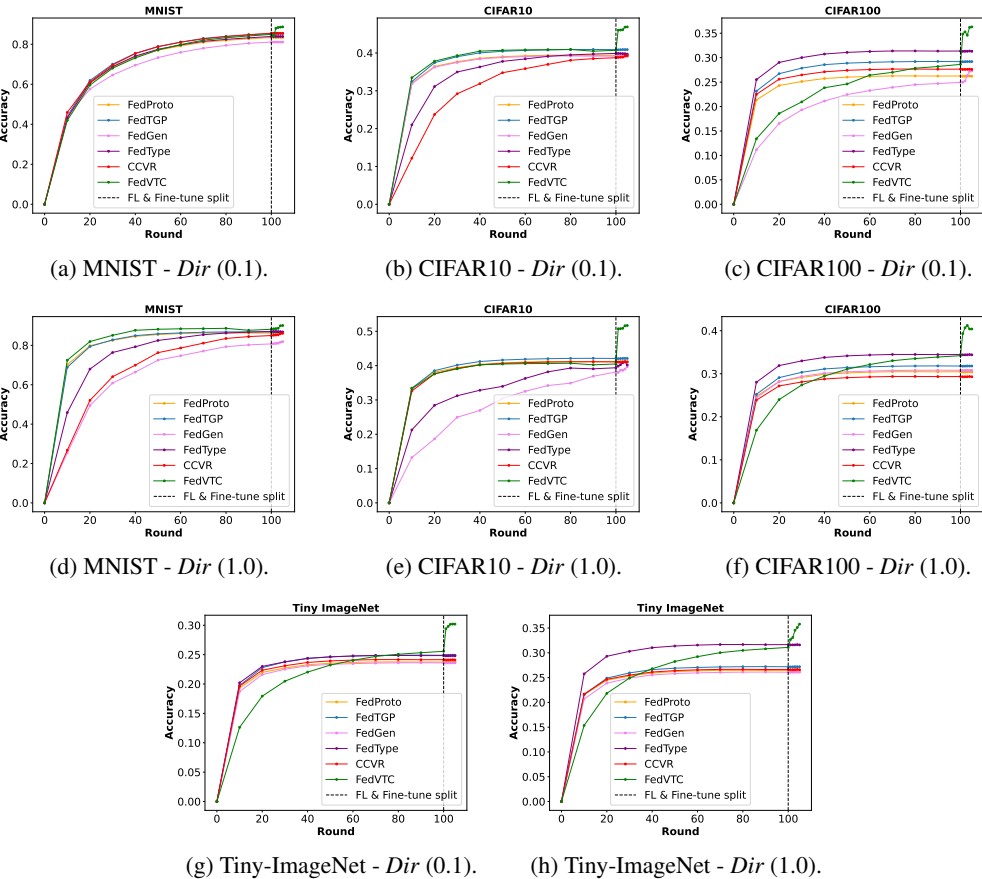

(a) MNIST - *Dir* (0.1).

(b) CIFAR10 - *Dir* (0.1).

(c) CIFAR100 - *Dir* (0.1).

(d) MNIST - *Dir* (1.0).

(e) CIFAR10 - *Dir* (1.0).

(f) CIFAR100 - *Dir* (1.0).

(g) Tiny-ImageNet - *Dir* (0.1).

(h) Tiny-ImageNet - *Dir* (1.0).

Figure 7: The per-round average generalization accuracy (i.e., the mean of each local model's accuracy on the global validation set). The vertical dashed line splits the standard FL iterations and the local fine-tuning procedure.

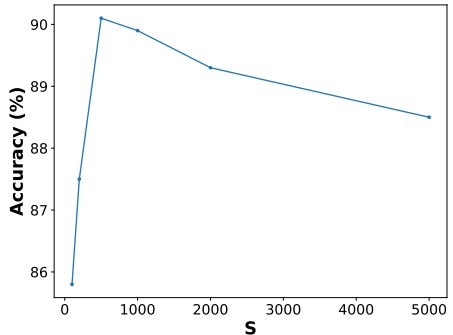 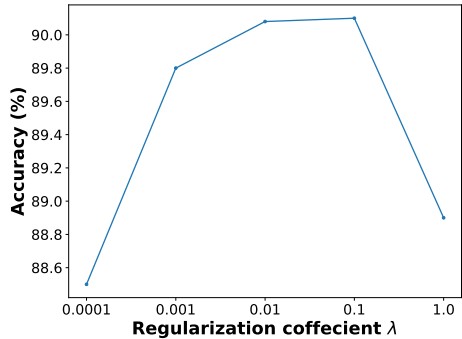

(a) Generalization accuracy on MNIST with $Dir(1.0)$ vs. the number of synthetic samples per class.

(b) Generalization accuracy on MNIST with $Dir(1.0)$ vs. the regularization coffecient $\lambda$.

Figure 8: The generalization accuracy with different values of $\lambda$.

| | Before fine-tuning | After Fine-tuning | Improvement |
|---|---|---|---|
| MNIST $Dir(0.1)$ | 84.9% | 88.7% | ↑ 3.8% |
| MNIST $Dir(1.0)$ | 86.8% | 90.1% | ↑ 3.3% |
| CIFAR10 $Dir(0.1)$ | 40.5% | 46.9% | ↑ 6.4% |
| CIFAR10 $Dir(1.0)$ | 41.6% | 51.7% | ↑ 10.1% |
| CIFAR100 $Dir(0.1)$ | 27.6% | 36.3% | ↑ 8.7% |
| CIFAR100 $Dir(1.0)$ | 32.8% | 40.4% | ↑ 7.6% |
| Tiny-ImageNet $Dir(0.1)$ | 25.1% | 30.2% | ↑ 5.1% |
| Tiny-ImageNet $Dir(1.0)$ | 29.8% | 35.8% | ↑ 6.0% |

Table 7: The generalization accuracies of FedVTC before and after fine-tuning with the synthetic samples.

### B.5 QUALITY OF SYNTHETIC SAMPLES

In this paper, the objective of generating synthetic samples is to produce training data for fine-tuning instead of generating realistic images. Therefore, traditional metrics measuring how generated images resemble real images (like FID score Heusel et al., 2017) do not accurately reflect the quality of the synthetic samples. As a replacement, we compare the accuracy of FedVTC before and after fine-tuning to examine the quality of the synthetic samples. As depicted in Figure 7 and Table 7, the accuracy of FedVTC significantly increases after fine-tuning, demonstrating a good quality of the synthetic samples.

## C USE OF LARGE LANGUAGE MODELS

Large Language Models are only used to correct spelling and grammatical mistakes in this paper.

