# OpenReview forum: "Bridging Generalization Gap of Heterogeneous Federated Clients Using Generative Models"
_ICLR.cc/2026/Conference — ICLR 2026 Poster_

### Official Review · Reviewer_fKa4 · 2025-10-28

**Soundness:** 2
**Presentation:** 2
**Contribution:** 2
**Rating:** 6
**Confidence:** 4

**Summary:**

This paper addresses the generalization challenge in model-heterogeneous federated learning where clients have different model architectures, making standard parameter aggregation infeasible. The proposed framework, FedVTC, avoids parameter sharing and instead has clients exchange feature distribution statistics (mean and covariance) with the server. Each client then trains a local variational transposed convolutional network, using latent variables sampled based on the aggregated statistics, to generate synthetic data. By fine-tuning their local models on this generated data, clients improve generalization without needing a public dataset.

**Strengths:**

The motivation of the method is described accurately and in detail, the algorithm process is also explained clearly, and the specific ablation experiments are very comprehensive.

**Weaknesses:**

1. Since each client needs to generate high-quality samples during training, please provide a more detailed analysis of the associated computational cost. Is this process computationally expensive, and if so, how does it affect training efficiency?

2. The paper claims to address model heterogeneity. Please clarify whether the proposed method can handle heterogeneous architectures (e.g., ResNet, MobileNet, CNNs) rather than merely different-sized variants of the same backbone (e.g., ResNet-18 vs. ResNet-50).

3. Since the primary goal of the proposed method is to improve generalization under heterogeneous model settings, the experiments should explicitly evaluate cross-domain generalization. In particular, results should be presented for each client model tested across all global test datasets. Currently, Figure 7 lacks details about the experimental setup, and Table 1 ambiguously refers to “all datasets and distributions.” Please describe these configurations more precisely to ensure reproducibility and clarity.

4. The comparative analysis omits several VAE-based federated learning methods, which are highly relevant to the proposed approach. Including these baselines would provide a more comprehensive evaluation of the method’s relative advantages.

5. In heterogeneous model scenarios, achieving generalization capability is likely related to class-specific feature representations. Even if data distributions differ across clients, a visualization (e.g., feature embeddings or attention maps) illustrating shared generalizable features would greatly enhance the interpretability of the results.

6. In the privacy validation experiments, the visual differences between reconstructed and original samples are evident to the human eye. However, for a machine learning system, such differences should be quantitatively evaluated (e.g., via PSNR, SSIM, or feature similarity metrics). A quantitative privacy comparison would make the evaluation more rigorous.

7. Several detail-related issues should be addressed. The abbreviation “TC” is undefined and should likely be “VTC”; please ensure consistency throughout the manuscript. Additionally, in line 195, “e.g.” should be written as “e.g.,” following standard academic style.

**Questions:**

Please refer to above

---

> ### Author Response · Authors · 2025-11-23
> **Rebuttal for W1-5**
>
> Dear reviewer,
>
> Thanks for your constructive feedback.
>
> We have carefully looked through every point of your review and refined the paper based on your comments.
>
> __For W1,__
>
> For computation overhead, we’d like you to refer to Table 5 in Appendix B. As shown in the Table, the VTC model is a lightweight model with only four transposed convolution layers. The number of parameters in a VTC model is very small, even compared with the smallest client model (ResNet18). Specifically, the parameter amount in a VTC model is about 0.5% of ResNet18. Therefore, the additional computation cost of training a VTC model is almost negligible. To verify this point, we’ve recorded the overall GPU training time (on CIFAR10) of all methods, shown in the table below:
>
> |                | GPU hours |
> |:--------------:|:---------:|
> |    FedProto    |    5.18   |
> |     FedTGP     |    5.21   |
> |      CCVR      |    5.53   |
> |     Fedtype    |    6.16   |
> |     FedGen     |    5.46   |
> | FedVTC (ours)  |    5.31   |
>
> Overall, the total GPU training time of FedVTC is 0.86~1.02$\times$ the baselines, which indicates no significant difference.
>
> __For W2,__
>
>
> We’ve conducted an additional evaluation round on MNIST-Dir(0.1). In the new round, we re-allocate clients into five uniform clusters with five architectures: {4-layer CNN, MobileNet, AlexNet, VGG-16, ResNet-18}:
>
> |               | Generalization Accuracy (%) on MNIST |
> |:-------------:|:------------------------------------:|
> |    FedProto   |                 83.2                 |
> |     FedTGP    |                 84.8                 |
> |     FedGen    |                 82.0                 |
> |      CCVR     |                 84.3                 |
> |    FedType    |                 84.1                 |
> |__ FedVTC (ours)__ |                 __88.9__                 |
>
>
> Experiment results show that FedVTC consistently outperforms the baselines, demonstrating a better generalization ability across different model architectures.
>
> __For W3,__
>
> We sincerely apologize for the confusion. As stated in the experiment setup (“Dataset” and “System implementation” in Section 4.1), each dataset contains a training set and a validation set, and we divide the training set over clients for FL training.
>
> Furthermore, as depicted in the “generalization accuracy” paragraph in Section 4.2, we evaluate each client’s model on the global validation set for generalization accuracy and record the average performance in Table 1. Therefore, your request “results should be presented for each client model tested across all global test datasets” is exactly what we are doing right now.
>
> For clarity, we have updated the descriptions of Table 1 and Figure 7:
>
> *Table 1. The average generalization accuracy (in %) of each client’s local model on the global validation dataset (with mean ± SD)*.
>
> *Figure 7. The per-round average generalization accuracy (i.e., the mean of each local model’s accuracy on the global validation set). The vertical dashed line splits the standard FL iterations and the local fine-tuning procedure.*
>
> __For W4,__
>
> We’ve evaluated FedVAE [1] on MNIST (Dir 0.1), where clients collaboratively train a global VAE, and use it to generate trajectory data to enhance local training.
>
>
> |               | Generalization Accuracy (%) on MNIST |
> |:-------------:|:------------------------------------:|
> |    FedVAE     |                 86.0                 |
> | FedVTC (ours) |                 88.7                 |
>
>
> As shown in the table above, FedVTC outperforms FedVAE with higher generalization accuracy. The possible reasons are: 1. Unlike VTC, FedVAE employs a linear decoder, which does not capture the non-linear transformation from the latent space to the synthetic data. 2. The feature distribution in FedVAE natively follows a standard Gaussian distribution with a mean of zero and a covariance of I. Comparatively, in FedVTC, the mean is set to global prototypes, and the standard deviation is optimized through gradient descent, which captures the feature distribution more precisely.
>
> __For W5,__
>
> We appreciate your suggestion. We have examined every possible pair of the 100 clients and evaluated the cosine similarity of their class-wise prototypes (mean of feature representation) on the MNIST dataset (Dir 1.0). The average cosine similarities for every pair are {0.683, 0.625, 0.711, 0.705, 0.687, 0.719, 0.688, 0.683, 0.692,0.678} for classes 0-9 respectively. This means that in FedVTC, clients tend to generate similar feature embeddings for all categories, despite their non-iid local data.

---

> ### Author Response · Authors · 2025-11-23
> **Rebuttal for W6-7**
>
> __For W6,__
>
> Following your advice, we have evaluated the SSIM score between the generated samples and every client’s original samples. The __maximum__ SSIM value is shown in the table below:
>
>
> |          | MNIST | CIFAR10 | CIFAR100 | Tiny ImageNet |
> |:--------:|:-----:|---------|----------|---------------|
> | Dir(0.1) |  0.39 | 0.36    | 0.21     | 0.19          |
> | Dir(1.0) |  0.4  | 0.32    | 0.29     | 0.16          |
>
> Specifically, “maximum SSIM” means that for any pair of generated image and original image, the similarity score will not exceed this value. As shown in the table, the maximum SSIM score remains low in all scenarios. This means that the context of synthetic images diverges drastically from real images, which helps our method preserve privacy.
>
> __For W7__,
>
> Thanks for your advice. We have removed the typo in the pdf file and replaced “TC” with “VTC” for consistency.
>
> Regards,
>
> [1] Yuchen Jiang, Ying Wu, Shiyao Zhang, and James J.Q. Yu. Fedvae: Trajectory privacy-preserving based on federated variational autoencoder. In IEEE 98th Vehicular Technology Conference (VTC2023-Fall), 2023.

---

### Official Review · Reviewer_MpGA · 2025-10-28

**Soundness:** 3
**Presentation:** 3
**Contribution:** 2
**Rating:** 6
**Confidence:** 3

**Summary:**

This paper addresses the challenge of data and model heterogeneity in federated learning (FL), where clients possess non-IID data and diverse model architectures, leading to limited generalization. The authors propose FedVTC, a model-heterogeneous FL framework that leverages variational transposed convolution (VTC) networks to generate synthetic data for fine-tuning local models. Instead of sharing model parameters, clients communicate feature distribution statistics (prototypes and standard deviations) with the server. The VTC model is trained using a composite loss combining evidence lower bound (ELBO) and distribution matching (DM) terms. Experiments on MNIST, CIFAR10, CIFAR100, and Tiny-ImageNet demonstrate improved generalization accuracy, reduced communication costs, and efficient memory usage compared to existing methods.

**Strengths:**

The use of variational transposed convolution to produce synthetic samples for fine-tuning local models is a distinctive approach, avoiding dependency on public datasets or parameter aggregation.

The method is evaluated on four benchmark datasets under varying non-IID settings (Dir(0.1) and Dir(1.0)), demonstrating superior generalization accuracy over five state-of-the-art baselines.

FedVTC achieves lower communication costs than FedProto, FedTGP, FedGen, CCVR, and FedType and maintains comparable or reduced memory usage through alternating training of local and VTC models.

The authors argue that transmitting prototypes and standard deviations does not expose raw data, supported by visual evidence of synthetic samples diverging from real images.

**Weaknesses:**

Although FedVTC is compared to five baselines, the discussion overlooks potential overlaps or distinctions with contemporary approaches such as those using diffusion models or zero-shot learning.

 Generalization performance is measured solely by accuracy; additional metrics like per-class precision/recall or robustness under extreme non-IID conditions would strengthen the claims.

The requirement for homogeneous VTC models for aggregation may limit applicability in fully heterogeneous settings where clients cannot support the same VTC structure.

The paper does not analyze cases where VTC fails to generate useful synthetic data (e.g., under high data skew or low client participation), nor does it explore the impact of varying latent dimensions on performance.

While memory usage is addressed, the additional computational cost of training VTC networks is not quantified or compared to baseline methods.

No formal convergence guarantees or analysis are provided for the alternating optimization of local models and VTC networks, which is critical for FL methods.

**Questions:**

Please respond to the Weaknesses.

---

> ### Author Response · Authors · 2025-11-23
> **Rebuttal for W1-4**
>
> Dear reviewer,
>
> Thanks for your constructive feedback.
>
> We have carefully looked through every point of your review and refined the paper based on your comments.
>
> __For W1__,
>
> We have updated the pdf file and added a new subsection in the literature review, discussing the limitations of existing federated diffusion/zero-shot frameworks:
>
> *2.3 FEDERATED TRAINING OF GENERATIVE MODELS*
>
> *Most existing FL schemes for training a generative model have several practical limitations. For instance, [1] and [2] propose to train a global diffusion model in FL networks. However, they require a public dataset to facilitate knowledge distillation among client models, which is not usually accessible. [3] trains a global generative adversarial network (GAN) by maximizing the discrepancy between client models. Clients use the downloaded GAN to produce training samples from unseen classes, thereby enhancing the performance of zero-shot prediction. However, the transmission of client models might cause overwhelming communication overhead. [4] utilizes a contrastive language-image pretraining (CLIP) model to guide clients’ local training for better few/zero-shot performance, while a pre-existing CLIP model is not usually available.*
>
> In summary, existing federated diffusion/zero-shot frameworks usually obtain several limitations, such as dependence on a public dataset or pre-trained model, and large communication costs. Comparatively, our method outperforms these works by abstaining from these limitations.
>
> __For W2,__
>
>
> We have evaluated the average per-class precision across every client on MNIST and CIFAR10 (Dir-1.0). The results are shown below:
>
> |            | Client Average Per-class Precision on MNIST |
> |------------|-----------------------------------------------|
> | 0   | 0.912                                         |
> | 1 | 0.881                                        |
> |2    | 0.919                                         |
> | 3       | 0.897                                        |
> |4      | 0.877                                         |
> |5        | 0.902                                        |
> | 6     | 0.897                                         |
> | 7    | 0.886                                         |
> | 8      | 0.910                                         |
> | 9    | 0.896                                        |
>
>
>
> |            | Client Average Per-class Precision on CIFAR10 |
> |------------|-----------------------------------------------|
> | airplane   | 0.565                                         |
> | automobile | 0.554                                         |
> | bird       | 0.543                                         |
> | cat        | 0.557                                         |
> | deer       | 0.576                                         |
> | dog        | 0.574                                         |
> | frog       | 0.564                                         |
> | horse      | 0.577                                         |
> | ship       | 0.571                                         |
> | truck      | 0.562                                         |
>
> As shown in the table, the precision has little variance. This means that the performance of FedVTC is well-balanced across all classes.
>
> __For W3,__
>
> In practice, the restriction of sharing a common structure for an additional model (e.g., a VTC model or a proxy model [5]) is much less than the fundamental local model. For FL clients, sharing a common structure of local models is not always applicable due to several reasons, like intellectual property, unique model ownership, and privacy concerns [6,7,8]. While for additional models, these restrictions seldom apply. Therefore, sharing the same VTC structure is usually feasible in heterogeneous settings.
>
> __For W4,__
>
> As depicted in the experiment setup (Section 4.1), data is allocated to clients following a Dirichlet (Dir) distribution with parameter 0.1. The less the Dir parameter is, the more skewed local data distributions are. According to existing works [9], a Dir parameter 0.1 usually indicates extremely unbalanced local data distributions. Besides, the client participation is set to only 10%. Therefore, the efficacy of FedVTC has already been verified in an environment with high data skew and low client participation.
>
> In classic FL, the dimension of the latent feature space is assumed to be fixed. The setting of varying latent dimensions refers to *vertical federated learning (VFL)* [11], which is beyond the scope of this paper.

---

> ### Author Response · Authors · 2025-11-23
> **Rebuttal for W5-6**
>
> __For W5__,
>
> For computation overhead, we’d like you to refer to Table 5 in Appendix B. As shown in the Table, the VTC model is a lightweight model with only four transposed convolution layers. The number of parameters in a VTC model is very small, even compared with the smallest client model (ResNet18). Specifically, the parameter amount in a VTC model is about 0.5% of ResNet18. Therefore, the additional computation cost of training a VTC model is almost negligible. To verify this point, we’ve recorded the overall GPU training time (on CIFAR10) of all methods, shown in the table below:
>
> |                | GPU hours |
> |:--------------:|:---------:|
> |    FedProto    |    5.18   |
> |     FedTGP     |    5.21   |
> |      CCVR      |    5.53   |
> |     Fedtype    |    6.16   |
> |     FedGen     |    5.46   |
> | FedVTC (ours)  |    5.31   |
>
> Overall, the total GPU training time of FedVTC is 0.86~1.02$\times$ the baselines, which indicates no significant difference.
>
> __For W6,__
>
> We would like you to understand that conducting a comprehensive convergence analysis of training a VTC/VAE model in FL is very complicated, which is beyond the scope of this paper. As a matter of fact, even the convergence property of a Centralized VAE has just been proposed recently in this year [10]. We will explore how to extend the existing theories in [10] to decentralized settings and derive a convergence analysis of FL in our future work.
>
> Regards,
>
> [1] Fiona Victoria Stanley Jothiraj and Afra Mashhadi. Phoenix: A federated generative diffusion model. In Companion Proceedings of the ACM Web Conference 2024, WWW ’24, pp. 1568–1577,
> New York, NY, USA, 2024. Association for Computing Machinery.
>
> [2] Zihao Peng, Xijun Wang, Shengbo Chen, Hong Rao, Cong Shen, and Jinpeng Jiang. Federated learning for diffusion models. IEEE Transactions on Cognitive Communications and Networking, 2025.
>
> [3] Lan Zhang, Dapeng Wu, and Xiaoyong Yuan. Fedzkt: Zero-shot knowledge transfer towards resource-constrained federated learning with heterogeneous on-device models. In 2022 IEEE 42nd International Conference on Distributed Computing Systems (ICDCS), pp. 928–938. IEEE, 2022.
>
> [4] Jiangming Shi, Shanshan Zheng, Xiangbo Yin, Yang Lu, Yuan Xie, and Yanyun Qu. Clip-guided federated learning on heterogeneous and long-tailed data. AAAI’24. AAAI Press, 2024.
>
> [5] Jiaqi Wang, Chenxu Zhao, Lingjuan Lyu, Quanzeng You, Mengdi Huai, and Fenglong Ma. Bridging model heterogeneity in federated learning via uncertainty-based asymmetrical reciprocity learning. In Proceedings of the 41st International Conference on Machine Learning, 2024.
>
> [6] Mang Ye, Xiuwen Fang, Bo Du, Pong C. Yuen, and Dacheng Tao, Heterogeneous Federated Learning: State-of-the-art and Research Challenges. ACM Comput. Surv. 56, 3, Article 79, 2023.
>
> [7] Zhang, J.; Gu, Z.; Jang, J.; Wu, H.; Stoecklin, M. P.; Huang,
> H.; and Molloy, I. Protecting intellectual property of deep neural networks with watermarking. In ASIA-CCS, 2018.
>
> [8] Li, Q.; Wen, Z.; Wu, Z.; Hu, S.; Wang, N.; Li, Y.; Liu, X.;
> and He, B. 2021a. A Survey on Federated Learning Systems: Vision, Hype and Reality for Data Privacy and Protection. IEEE Transactions on Knowledge and Data Engineering.
>
> [9] Mi Luo, Fei Chen, Dapeng Hu, Yifan Zhang, Jian Liang, and Jiashi Feng, No fear of heterogeneity: Classifier calibration for federated learning with non-IID data. In Proceedings of the 35th Conference on Neural Information Processing Systems, 2021.
>
> [10] Surendran, Sobihan, Antoine Godichon-Baggioni, and Sylvain Le Corff, Theoretical Convergence Guarantees for Variational Autoencoders”, AISTATS, 2025.
>
> [11] Y. Liu et al., "Vertical Federated Learning: Concepts, Advances, and Challenges," in IEEE Transactions on Knowledge and Data Engineering, vol. 36, no. 7, pp. 3615-3634, July 2024.

---

### Official Review · Reviewer_xMVi · 2025-10-28

**Soundness:** 3
**Presentation:** 2
**Contribution:** 2
**Rating:** 4
**Confidence:** 3

**Summary:**

This paper proposes FedVTC, a model-heterogeneous federated learning framework based on Variational Transposed Convolution. The framework generates synthetic data to fine-tune local models, thereby improving the generalization of heterogeneous clients without relying on public datasets. The paper compares FedVTC with several existing methods, but it lacks comparisons against strong works from 2025.

**Strengths:**

1. The paper proposes using FedVTC to generate synthetic data, which can be used to fine-tune local models and improve their generalization ability, thereby eliminating the reliance on public datasets.

2. FedVTC avoids exposure of raw data by transmitting prototypes, covariances, and VTC models, thus preventing additional privacy risks.

3. A new objective function is designed for training the VTC model, which includes the standard negative ELBO loss and a distribution matching (DM) loss, regularizing the training process and improving the quality of the generated synthetic data.

**Weaknesses:**

1. Some symbols in the formulas are not defined, such as the symbol $V_i$ in Formula 5, which lacks a definition.

2.  The proposed FedVTC framework does not introduce an entirely new solution or framework, but rather builds upon existing methods, which results in a somewhat limited level of innovation.

3. The comparative experiments seem to only compare with works published before 2025, without including comparisons with excellent works from 2025.

4. The flowchart in the paper is somewhat simplistic. For example, in Figure 1, the elements, colors, and arrows are used in a rather basic way, and the relationships and operational details between the steps are not fully demonstrated.

**Questions:**

1. I would like to know the meaning of the symbol $V_i$ in Formula 5.

2. Could you add experiments comparing with the outstanding works from 2025 to provide a clearer comparison of experimental results?

---

> ### Author Response · Authors · 2025-11-23
> **Rebuttal for W1 and Q1**
>
> Dear reviewer,
>
> Thanks for your constructive feedback.
>
> We have carefully looked through every point of your review and refined the paper based on your comments.
>
> __For W1 and Q1,__
>
> $\boldsymbol{v}_{i}$ is equivalent to $\boldsymbol{v}$ defined in Section 3.2 and Figure 1. $i$ is simply the batch index, which can be omitted for brevity.
>
> As shown in Figure 1 and Section 3.2, $\boldsymbol{v}=\boldsymbol{z}+\boldsymbol{\sigma}_{k}\odot \boldsymbol{\epsilon}$.
>
> $\boldsymbol{\sigma}_{k}$ is the standard deviation of the latent variables of client k, which is a learnable variable. $\boldsymbol{\epsilon}$ is a standard Gaussian variable following distribution $\mathcal{N}(\boldsymbol{0}, \boldsymbol{I})$, and “$\odot$” stands for element-wise product.
>
> In FedVTC. each client k’s latent variable is assumed to be a Bayesian random variable with distribution $\mathcal{N}(\boldsymbol{z}, Σk)$. The elements in the diagonal covariance matrix $Σk$ are learnable parameters. To update $Σk$, client k needs to sample a variable from $\mathcal{N}(\boldsymbol{z}, Σk)$, and feed the variable forward to the loss function in Equation (4).
>
> __However, the sampling process is discrete and non-differentiable, which makes it prohibitive to update $Σk$ through gradient descent. Therefore, the well-known reparametrization trick is applied.__ Instead of sampling from  $\mathcal{N}(\boldsymbol{z}, Σk)$, we keep $\boldsymbol{z}$ fixed, and introduce $\boldsymbol{v}=\boldsymbol{z}+\boldsymbol{\sigma}_{k}\odot \boldsymbol{\epsilon}$.
>
> In this case, $\boldsymbol{\sigma}_{k}$ equals the diagonal elements in the covariance matrix $Σk$.
>
> Within $\boldsymbol{v}$, we only sample from the Gaussian noise $\boldsymbol{\epsilon} \sim \mathcal{N}(0, \boldsymbol{I})$. Consequently, $\boldsymbol{v} \sim \mathcal{N}(\boldsymbol{z}, Σk)$. By feeding $\boldsymbol{v}$ forward, the loss is differentiable with respect to $\sigma_{k}$, enabling us to update $\sigma_{k}$ through gradient descent, which is equivalent to updating $Σk$.
>
> The specific details of the reparameterization trick can be found in Section 2.4 of [1].

---

> ### Author Response · Authors · 2025-11-23
> **Rebuttal for Q2 and W3**
>
> We’ve implemented the state-of-the-art pFedAFM framework (ICDE 2025) in paper [2], where clients employ a proxy feature embedding model for knowledge distillation. As shown in the table below, our method consistently outperforms pFedAFM with higher generalization accuracy.
>
> |                        |   pFedAFM  | FedVTC (ours) |
> |:----------------------:|:---------:|---------------|
> |     MNIST Dir(0.1)     | 85.6(0.5) | 88.7(0.7)     |
> |     MNIST Dir(1.0)     | 87.2(0.4) | 90.1(0.1)     |
> | CIFAR10 Dir(0.1)       | 38.9(0.9) | 46.9(0.6)     |
> | CIFAR10 Dir(1.0)       | 40.5(0.5) | 51.7(0.7)     |
> | CIFAR100 Dir(0.1)      | 32.2(1.0) | 36.3(0.8)     |
> | Cifar100 Dir(1.0)      | 34.9(0.6) | 40.4(0.3)     |
> | Tiny ImageNet Dir(0.1) | 24.2(0.8) | 30.2(0.3)     |
> | Tiny ImageNet Dir(1.0) | 27.3(0.2) | 35.8(0.5)     |
>
>
> Meanwhile, FedVTC obtains lower communication costs (GB) than pFedAFM:
>
> |               | pFedAFM | FedVTC |
> |:-------------:|:-------:|--------|
> |     MNIST     |   1.94  |  0.70  |
> |    CIFAR10    |   5.88  |  2.90  |
> |    CIFAR100   |  41.27  |  26.49 |
> | Tiny ImageNet |  147.10 |  52.71 |
>
>
>
> Besides, if you have any particular methods you wish us to add, please tell us, and we will be happy to implement them.

---

> ### Author Response · Authors · 2025-11-23
> **Rebuttal for W2**
>
> We acknowledge that the design of FedVTC is motivated by the variational autoencoder (VAE) framework [1]. __However, instead of applying VAE to FL directly, we make several innovative adaptations to make FedVTC fit in FL settings.__ First, considering the heterogeneous architecture of feature extractors (equivalent to encoders in VAE) among clients, we discard the symmetric architecture (encoder & decoder), and merely incorporate the decoder (i.e., VTC) module into FL clients, to ensure a smooth deployment of FedVTC in model-heterogeneous FL settings. Second, in the original VAE method, the feature space is assumed to follow a standard Gaussian distribution with a zero mean. In FedVTC, considering the divergence of class-wise prototypes, the mean of the feature space is set to the class-wise global prototypes (see Section 3.2). This design enables FedVTC to learn distinct feature representations between different classes, resulting in better performance.
>
> __Furthermore, using existing techniques for synthetic image generation is still regarded as a significant contribution in FL, like [3] and [4]. Besides, FedVTC exhibits several advantages over them. Specifically__:
>
> [3] uses a generative adversarial network (GAN) to generate images. The GAN is trained on the server by maximizing the discrepancy between client models. This module requests the transmission of clients’ large-scale local models, which is communicationally expensive.
>
> [4]generates synthetic images using dataset condensation, which requires a pre-trained data encoder. In practice, such a pre-trained encoder is not usually available.
>
> In comparison, FedVTC is free of these limitations, demonstrating better efficiency and feasibility.
>
> __Additionally, the original VAE framework is trained with ELBO sole, while we propose a novel objective function to train VTC by combining ELBO with the distribution matching loss. Results in Table 4 show that this module achieves higher accuracy.__

---

> ### Author Response · Authors · 2025-11-23
> **Rebuttal for W4**
>
> We sincerely apologize for the ambiguity.
>
> We have changed the use of colors in Figure 1, with different colors indicating different operations. Specifically, __black arrows indicate the fundamental forward propagation, red arrows indicate reparameterization, and blue errors indicate upsampling. Please see the latest pdf file for details.__
>
> We have also added explanations of the arrows in the text of Section 3.2:
>
> "
>
> As Figure 1(b) shows, aside from the fundamental local model fk, FedVTC introduces a variational transposed convolution (VTC) model ψk : R p → R d to each client k. VTC is an upsampling network that takes a latent variable z ∈ R p as input to produce an enlarged data sample x ′ ∈ R d (Dumoulin & Visin, 2016). Similar to a VAE (Kingma & Welling, 2013), for each sample x with the corresponding z, client k samples a random variable v from the distribution N (v|z, Σk), and forwards v to ψk to derive x ′ (__blue arrow in Figure 1(b)__). The covariance matrix Σk ∈ R p×p can be learned via the gradient method. To make Σk differentiable, FedVTC applies the well-known reparametrization trick as per VAE and lets v = z + σk ⊙ ϵ (__red arrow in Figure 1(b)__). ϵ ∈ R p is a random Gaussian noise with distribution ϵ ∼ N (ϵ|0, I), σk = [σ1, σ2, ......, σp] ⊤ indicates the reparameterized standard deviation (SD), and ”⊙” represents element-wise product. In this case, for each entry in Σk, we have (Σk)ii = σ 2 i and (Σk)ij = 0 for any i ̸= j, and each σi (1 ≤ i ≤ p) can be learned through gradient descent.
>
> "
>
> Regards,
>
> [1] "Auto-Encoding Variational Bayes", arXiv:1312.6114.
>
> [2] “pFedAFM: Adaptive Feature Mixture for Data-Level Personalization in Heterogeneous Federated Learning on Mobile Edge Devices”, ICDE 2025.
>
> [3] “Fedzkt: Zero-shot knowledge transfer towards resource-constrained federated learning with heterogeneous on-device models”. In 2022 IEEE 42nd International Conference on Distributed Computing Systems (ICDCS).
>
> [4] “Overcoming Data and Model heterogeneities in Decentralized Federated Learning via Synthetic Anchors”, ICML, 2024.

---

### Official Review · Reviewer_Ltp9 · 2025-11-01

**Soundness:** 3
**Presentation:** 3
**Contribution:** 3
**Rating:** 6
**Confidence:** 4

**Summary:**

This paper proposes a method to improve the generalization of personalized Federated Learning, specifically enhancing accuracy on unseen data. The authors suggest generating synthetic data at the client side to fine-tune the global model after the federated training phase. This approach improves the performance of local models without requiring access to a public global dataset. To achieve this, the paper introduces a variational transposed convolutional neural network for synthetic data generation. Experimental results demonstrate significant improvements in the average client accuracy across multiple settings, including four datasets and two non-IID scenarios.

**Strengths:**

S1. The paper is clear to understand and follow
S2. The approach of fine-tuning the local models of clients using synthetic data is not new but the method using to generate the synthetic data is novel.
S3. Significant improvement in average accuracy among clients are shown.

**Weaknesses:**

W1. Computation and Communication Overhead: The proposed method requires the server to train and distribute the VTC model to clients. In addition, clients must generate synthetic data locally. These steps introduce non-trivial computational overhead compared to standard federated learning. Please discuss the communication and computation requirements of the proposed method and compare them with those of other existing approaches.

W2. Privacy Preservation: Although the authors discuss privacy preservation in the paper, the examples shown in Figure 4 do not convincingly demonstrate that data privacy is maintained. It would be beneficial to quantitatively assess the similarity between the original samples and the corresponding generated data. For instance, the authors could use the method proposed in Z. Wang, A. C. Bovik, H. R. Sheikh, and E. P. Simoncelli, “Image quality assessment: from error visibility to structural similarity,” IEEE Transactions on Image Processing, vol. 13, no. 4, pp. 600–612, 2004, which introduces the Structural Similarity Index (SSIM).

W3. The authors compares FedVTC with 5 other state of the art. However, the authors should discuss the approach in this paper "Enhancing the Generalization of Personalized Federated Learning with Multi-head Model and Ensemble Voting" (https://ieeexplore.ieee.org/abstract/document/10579121) which also try to improve the generalization of Personalized Federated Learning.  It will be better if the author could show that proposed method outperform such kind of approach.

**Questions:**

See above

---

> ### Author Response · Authors · 2025-11-23
> **Rebuttal**
>
> Dear reviewer,
>
> Thanks for your constructive feedback.
>
> We have carefully looked through every point of your review and refined the paper based on your comments.
>
> __For W1,__
>
>
> For computation overhead, we’d like you to refer to Table 5 in Appendix B. As shown in the Table, the VTC model is a lightweight model with only four transposed convolution layers. The number of parameters in a VTC model is very small, even compared with the smallest client model (ResNet18). Specifically, the parameter amount in a VTC model is about 0.5% of ResNet18. Therefore, the additional computation costs of training a VTC model, and generating synthetic images using the VTC model, are almost negligible. To verify this point, we’ve recorded the overall GPU training time (on CIFAR10) of all methods, shown in the table below:
>
> |                | GPU hours |
> |:--------------:|:---------:|
> |    FedProto    |    5.18   |
> |     FedTGP     |    5.21   |
> |      CCVR      |    5.53   |
> |     Fedtype    |    6.16   |
> |     FedGen     |    5.46   |
> | FedVTC (ours)  |    5.31   |
>
> Overall, the total GPU training time of FedVTC is 0.86~1.02$\times$ the baselines, which indicates no significant difference.
>
> For communication overhead, please refer to Table 2 in the paper. We have already shown that FedVTC achieves the minimal communication cost among all methods.
>
> __For W2,__
>
> Following your advice, we have evaluated the SSIM score between the generated samples and every client’s original samples. The __maximum__ SSIM value is shown in the table below:
>
>
> |          | MNIST | CIFAR10 | CIFAR100 | Tiny ImageNet |
> |:--------:|:-----:|---------|----------|---------------|
> | Dir(0.1) |  0.39 | 0.36    | 0.21     | 0.19          |
> | Dir(1.0) |  0.4  | 0.32    | 0.29     | 0.16          |
>
> Specifically, “maximum SSIM” means that for any pair of generated image and original image, the similarity score will not exceed this value. As shown in the table, the maximum SSIM score remains low in all scenarios. This means that the context of synthetic images diverges drastically from real images, which helps our method preserve privacy.
>
> __For W3,__
>
> We have added the discussion of the pFedGen work to the literature review. Please have a look at Section 2.1 of the latest pdf submission:
>
> “Weight modification methods enhance the generalization of the global model by deducting the proportion of biased clients participating in global training……***pFedGen enables unseen clients to generate representation vectors using a feature extractor trained by previous clients, and only interacts with clients with the smallest representation distance. Consequently, these works develop a selective client sampling strategy that selects unimportant entities with lower possibilities to mitigate the negative effect of biased clients. Weight modification methods enhance the generalization of the global model by deducting the proportion of biased clients participating in global training. ”***
>
>
> Furthermore, we find that the proposed pFedGen method requires a common architecture for all clients’ base (i.e., feature extractor) layers, which is unsuitable in model-heterogeneous FL settings.
>
> Regards,

---

### Author Response · Authors · 2025-11-27
**Please have a look of our rebuttal**

Dear reviewers,

It has been several days since we posted our rebuttal. Given that there isn't much time left before the discussion deadline, we sincerely ask you to read through our rebuttal and give us a response.

Thanks for your time!

Regards

---

### Author Response · Authors · 2025-12-04
**Rebuttal Summary**

Dear Program Chairs, Senior Area Chairs, and Area Chairs,

We are presenting a rebuttal summary to help you progress the work better.

Our submission received four initial ratings of 6,6,6,4. The reviewers’ comments and our rebuttal are summarized as below:

### __Reviewer Ltp9 (rating: 6)__

The reviewer asked us to provide:
1. The analysis of computation/communication overhead,
2. The SSIM score between synthetic and raw images,
3. Discussion of additional literature.

We have included these contents comprehensively in the rebuttal.

### __Reviewer xMVi (rating: 4)__

The reviewer questioned the novelty of our paper. In the rebuttal, we concretely clarified the innovation of our method, such as the distinction between VTC & VAE and a novel objective function for VTC training.

The reviewer asked us to add some newest baselines in 2025 without giving any particular example. Correspondingly, we implemented the pFedAFM framework in the paper “pFedAFM: Adaptive Feature Mixture for Data-Level Personalization in Heterogeneous Federated Learning on Mobile Edge Device” (ICDE 2025).

Besides, the reviewer asked us to explain the meaning of a mathematical notation and refine the layout of Figure 1. We have addressed these points thoroughly in the rebuttal.

### __Reviewer MpGA (rating: 6)__

The reviewer asked us to provide additional contents, such as: discussion of existing diffusion/zero-shot works, per-class precision, robustness on highly-skewed data, and computation overhead. We have added these points comprehensively in the rebuttal.

The reviewer questioned the setting of fixed latent dimension/VTC architecture, and lack of convergence analysis. In the rebuttal, we specifically clarify that these points are beyond the scope of our paper.



### __Reviewer fKa4 (rating: 6)__

The reviewer mainly asked us to provide additional contents, such as:

1. Computation overhead,

2. Results on more model architectures,

3. Comparison between the FedVAE baseline,
4. SSIM score between synthetic and raw images.

We have included all points above in the rebuttal. Besides, we’ve removed some ambiguous expressions and typos in the paper following the reviewer’s advice.

Although we did not receive any further comments or ratings update from the reviewers, we are confident that all of the reviewers’ comments have been successfully addressed.

For more details, please have a look at our rebuttal below.

Thanks for your time.

Regards

---

### Meta-Review · Area_Chair_r7Uw · 2026-01-04

**Summary:**

This paper proposes FedVTC, a model-heterogeneous federated learning framework that improves generalization by having clients share feature distribution statistics rather than model parameters. Each client uses these statistics to train a lightweight Variational Transposed Convolution network that generates synthetic data for local fine-tuning. The method claims advantages in generalization accuracy, communication efficiency, and privacy preservation, validated across four datasets under strong non-IID settings. The approach avoids reliance on public datasets or shared model architectures, which are common limitations in prior heterogeneous FL methods.

**Reviewer Concerns:**

Reviewer Ltp9 (rating: 6): Raised concerns about computational/communication overhead, quantitative privacy evaluation (SSIM), and related work. The authors provided GPU hour comparisons, SSIM scores, and added literature discussion.
Reviewer xMVi (rating: 4): Questioned novelty, notation clarity, and lack of 2025 baselines. While the authors implemented pFedAFM and clarified notation via reparameterization, the core concern, limited novelty beyond adapting VAE concepts to FL, was not convincingly rebutted. The claimed “innovative objective function” and architectural choices remain incremental.
Reviewer MpGA (rating: 6): Requested discussion of diffusion/zero-shot FL methods, per-class metrics, robustness, and convergence analysis. The authors added relevant literature, per-class precision tables, and GPU hour data, but convergence analysis was explicitly deferred to future work.
Reviewer fKa4 (rating: 6): Asked for clarification on architecture heterogeneity, cross-dataset evaluation protocol, comparison, and quantitative privacy metrics. The authors responded with additional experiments across diverse architectures, SSIM scores.

**Reviewer Scores:**

I believe all reviewers will maintain their original scores, as they have already provided generally positive feedback. Regarding the reviewer who gave a score of 4, I think the authors did not directly address the reviewer's core concerns, so I expect that reviewer to keep their original score.

---

### Decision · Program_Chairs · 2026-01-26

Accept (Poster)